# Updates in Diagnostic Imaging for Infectious Keratitis: A Review

**DOI:** 10.3390/diagnostics13213358

**Published:** 2023-10-31

**Authors:** Maria Cabrera-Aguas, Stephanie L Watson

**Affiliations:** 1Save Sight Institute, Discipline of Ophthalmology, Faculty of Medicine and Health, The University of Sydney, Sydney, NSW 2000, Australia; stephanie.watson@sydney.edu.au; 2Sydney Eye Hospital, Sydney, NSW 2000, Australia

**Keywords:** infectious keratitis, corneal imaging, in vivo confocal microscopy, optical coherence tomography, artificial intelligence, deep learning, microbial keratitis

## Abstract

Infectious keratitis (IK) is among the top five leading causes of blindness globally. Early diagnosis is needed to guide appropriate therapy to avoid complications such as vision impairment and blindness. Slit lamp microscopy and culture of corneal scrapes are key to diagnosing IK. Slit lamp photography was transformed when digital cameras and smartphones were invented. The digital camera or smartphone camera sensor’s resolution, the resolution of the slit lamp and the focal length of the smartphone camera system are key to a high-quality slit lamp image. Alternative diagnostic tools include imaging, such as optical coherence tomography (OCT) and in vivo confocal microscopy (IVCM). OCT’s advantage is its ability to accurately determine the depth and extent of the corneal ulceration, infiltrates and haze, therefore characterizing the severity and progression of the infection. However, OCT is not a preferred choice in the diagnostic tool package for infectious keratitis. Rather, IVCM is a great aid in the diagnosis of fungal and Acanthamoeba keratitis with overall sensitivities of 66–74% and 80–100% and specificity of 78–100% and 84–100%, respectively. Recently, deep learning (DL) models have been shown to be promising aids for the diagnosis of IK via image recognition. Most of the studies that have developed DL models to diagnose the different types of IK have utilised slit lamp photographs. Some studies have used extremely efficient single convolutional neural network algorithms to train their models, and others used ensemble approaches with variable results. Limitations of DL models include the need for large image datasets to train the models, the difficulty in finding special features of the different types of IK, the imbalance of training models, the lack of image protocols and misclassification bias, which need to be overcome to apply these models into real-world settings. Newer artificial intelligence technology that generates synthetic data, such as generative adversarial networks, may assist in overcoming some of these limitations of CNN models.

## 1. Introduction

The cornea is essential for vision and contributes three-quarters of the eye’s refractive power. While cataracts in the developing world and age-related macular degeneration (in older patients) in the developed world are recognised as the leading causes of visual impairment, corneal blindness affects all ages and is a leading cause of irreversible visual impairment [1]. Despite continuous efforts to combat this disease, infectious corneal ulceration (infectious keratitis) still receives insufficient global attention [1]. Corneal ulceration (opacity) is among the top five causes of blindness and vision impairment globally [1,2,3]. In 2020, it was reported that 2.096 million people over 40 years of age suffered from blindness, and 3.372 million people have moderate to severe vision impairment from non-trachomatous corneal opacity [2].

Infectious keratitis is an ocular emergency requiring prompt attention as it can progress very rapidly, leading to severe complications, such as losing eyesight or even the eye [4]. Infectious keratitis is diagnosed using the patient’s history, clinical examination under a slit lamp and the microbiology results from staining and culture of the scrapings from the corneal ulcer [4,5]. The key to making the diagnosis is the identification of key features on slit lamp examination with fluorescein staining [6,7] (Figure 1). Along with aiding with diagnosis, slit lamp biomicroscopy is used in determining the severity of the infection [8].

Infectious keratitis is most often caused by bacteria. Other important causal organisms include viruses, fungi and parasites [4]. Bacterial keratitis is mostly caused by *Staphylococci* spp., *Pseudomonas aeruginosa* and *Streptococcus pneumoniae*. Patients generally manifest with a red eye, discharge, an epithelial defect, corneal infiltrates and sometimes hypopyon [9] (Table 1). Viral keratitis is commonly caused by herpes simplex virus (HSV). The type of HSV keratitis is determined based on the clinical features observed on the slit lamp examination. A dendritic or geographic ulcer is found in ‘epithelial HSV keratitis’. Stromal haze with or without ulcers, lipid keratopathy, stromal oedema, scarring, corneal thinning or vascularisation are found in ‘stromal HSV keratitis’. Stromal oedema and keratic precipitates are found in endothelial HSV keratitis. Stromal oedema, keratic precipitates and anterior chamber cells are found in ‘HSV keratouveitis’ [10,11,12] (Table 1).

Fungal keratitis is caused by filamentous (*Fusarium* spp., *Aspergillus* spp.) or yeast (*Candida* spp.) fungi. Clinical findings may include a corneal ulcer with irregular feathery margins, elevated borders, dry, rough texture, satellite lesions, Descemet’s folds, hypopyon, ring infiltrate, endothelial plaque, anterior chamber cells and keratic precipitates [13,14,15] (Table 1). Parasitic keratitis caused by *Acanthamoeba* spp. is a usual cause of infectious keratitis, which is a typically chronic and progressive condition. A unilateral or paracentral corneal ulcer with a ring infiltrate is commonly seen in patients with this infection. Patients may present with eyelid ptosis, conjunctival hyphemia, and pseudodendrites. As the infection progresses, deep stromal infiltrates, corneal perforation, satellite lesions, scleritis, and anterior uveitis with hypopyon may be found. Symptoms may include severe eye pain, decreased vision, foreign body sensation, photophobia, tearing and discharge [16,17,18] (Table 1).

The culture of corneal scrapings is the gold standard for diagnosis and to identify and isolate the causal organism of the infection [4,5,19]. However, the positive rate of such cultures ranges from 38–66% [4]. Antimicrobial resistance testing is routinely performed on bacterial isolates, with the results typically available after 48 h. Empiric antimicrobial therapy is commenced whilst awaiting these results to prevent a severe complication [4]. Due to the limitations of corneal scraping in recent years, a range of imaging techniques have been used to aid the diagnosis of and severity grading of infectious keratitis. Understanding the current status of corneal imaging for keratitis will be of benefit to clinicians in practice and researchers in the field. Imaging technologies may enable early diagnosis of the different types of infectious keratitis. The aim of this review was to describe the corneal imaging diagnostic tests in use for infectious keratitis and to discuss the progress of artificial intelligence in diagnosing and differentiating the different types of infectious keratitis.

## 2. Slit Lamp Biomicroscopy

The slit lamp is a stereoscopic biomicroscope which produces a focused beam of light with different heights, widths and angles to visualise and measure the anatomy of the adnexa and anterior segment of the eye [20]. The slit lamp is essential for the examination and diagnosis of patients with infectious keratitis [4]. Slit lamp photography started in the late 1950s, but the arrival of digital cameras in the 2000s substantially facilitated its use in ophthalmology [21]. There are two types of digital cameras: single lens reflex (SLR) or ‘point-and-shoot’. The choice of either type of camera for use in clinics depends on the budget, ease of use, photographic requirements, and ability of the user. SLRs are heavier, bulkier, and more costly than ‘point-to-view’ cameras. Another key feature to consider in selecting a camera is the megapixel resolution. One megapixel is equivalent to 1 million pixels. A photograph taken at 6 megapixels can be printed up to 11 inches (28 cm) × 14 inches (35.5 cm) without ‘pixelation’ (visible pixels). A 3.2-megapixel camera can meet the needs of clinical photography [22]. Other important features to consider include whether the camera has a macro mode for close-up photography, flash mode to light a dark scene or the object to be photographed, additional flash fixtures to create diffuse illumination, and ‘image stabilization’ or ‘vibration reduction’ technology to minimise camera shake in order to avoid blurred photographs [22].

An alternative to digital cameras is the ‘smartphone’, which was released in the late 2000s. A smartphone is a mobile phone that has the technology to run many advanced applications. In ophthalmology, such applications may include patient and physician education tools, testing tools, and photography [23]. Newer smartphones have rear camera resolution of up to 50 megapixels with image sensors, lens correction and optical plus electronic image stabilisation [24]. Due to the difficulty of holding a smartphone while operating the slit lamp, adapters to mount smartphones have been developed [21]. Limitations with the use of adapters include that they are not universal; that is, they are designed for certain smartphones or slit lamp models, and when they are attached to the slit lamp, binocular operation of the slit lamp is not feasible.

To overcome these issues, Muth et al. evaluated a new adapter that could be mounted in any smartphone or slit lamp and easily moved aside to enable binocular use of the slit lamp [21]. The images taken with the smartphone had an overall high quality and were as equally as good as the images taken with a slit lamp camera [21]. Currently, smartphones have built-in cameras appropriate for slit lamp imaging. If the smartphone is placed and handled adequately, the slit lamp image quality depends on three factors: the smartphone camera sensor’s resolution, the resolution of the slit lamp or microscope and the focal length of the smartphone camera system. The final image result depends on the smartphone’s software settings including autofocus, shutter speed, and internal post-processing algorithms when a compressed image format is used (i.e: .jpg). Newer smartphones that can take images in raw format will need more software-based post-processing [21,25].

## 3. Optical Coherence Tomography

In 1994, it was shown that the anterior chamber could be imaged using optical coherence tomography (OCT) with the same frequency (830 nm) used in posterior chamber imaging [7,26]. This advance enabled imaging and analysis of structures, such as the cornea and the anterior chamber angle. The predominant market for OCT applications has been for retinal imaging. The first commercial anterior segment devices, known as anterior segment optical coherence tomography (AS-OCT) that became commercially available were modified posterior chamber devices. Some of these AS-OCT devices were stand-alone, and others required modifications, such as an additional lens to modify the posterior chamber OCT devices. Modifications in 2001 to the light source and lens of the AS-OCT enabled higher frequency waves (1310 nm) to allow a higher resolution of the image and more precise measurement [7,27].

AS-OCT is now a well-recognised method for imaging the cornea and is often used in anterior chamber angle imaging for glaucoma [7,19]. The advantage of the AS-OCT is its ability to accurately measure the depth and width of the corneal ulcer, infiltrates, and haze to monitor the progress of corneal pathologies, such as superficial and deep infectious keratitis [19]. Other applications of the AS-OCT include the ability to measure the corneal thickness to determine the risk of corneal perforation, prediction of corneal cross-linking (PACK-CXL) response in infectious keratitis and highlighting corneal interface pathologies, such as interface infectious keratitis following lamellar keratoplasty (hyper-reflective band at the graft-host interface) [28], post-LASIK epithelial ingrowth (flap-host interface) [29] and valvular and direct non-traumatic corneal perforations associated with infectious keratitis [19,30]. With the increasing popularity of Small Incision Lenticule Extraction (SMILE) as a refractive procedure, AS-OCT will have a role in identifying and defining interface infections [31,32]. For bacterial keratitis, AS-OCT is currently an auxiliary tool to assist in diagnosis. Intra-operative AS-OCT is also emerging as a useful tool and may have applications in surgery for infectious keratitis by enabling delineation of involved structures and identifying corneal thinning (Figure 2).

### 3.1. Types of AS-OCT

Optical coherence devices in clinical use can be classified according to the type of image sampling as Time domain, Spectral Domain, and Swept Source. Each class of OCT has a different sampling speed and resolution, which has implications for the application of AS-OCT.

#### 3.1.1. Spectral Domain OCT

Spectral-domain (SD) AS-OCT assesses the frequency spectrum of the interference between a stationary reference mirror and the reflected light. Spatial and structural information are measured at the same time at all echo time delays (axial pixels). The advantage of a concurrent evaluation of all axial-depth scan (A-scan) pixels is that it enables an increase in scanning speeds of up to 100,000 A-scans/s with commercial devices and up to 20.8 million A-scans/s with research devices [33]. A higher resolution, up to 2 microns, had been achieved via a broader spectrum light source with SDAS-OCT [7,34]. SDAS-OCT has been used to assess corneal pathology, such as scarring and thinning (Figure 3A) and measure corneal thickness, including epithelial layer thickness (Figure 3B).

The use of SD AS-OCT in infectious keratitis has been reported in one cross-sectional study of 22 eyes by Soliman et al. [35], one case series with four eyes by Yamazaki et al. [7,36] and one cross-sectional study of 25 patients by Oliveira et al. [37]. Using SD-OCT (RTVue-100; Optovue, Freemont, CA, USA), Soliman et al. were able to distinguish infiltrates from scars, potentially allowing for the distinction of different stages of infectious keratitis as well as potentially identifying non-infective causes [7,19,35]. They found that although both infiltrates and scars were hyper-reflective, corneal infiltrates (Figure 1) had overlying defects or ‘opaque epithelial layers’ in combination with poorly defined and rounder borders. In contrast, scars had well-defined edges with a defect-free epithelium. In addition, cystic spaces were identified in some active lesions, which were correlated to areas of necrosis. By combining this classification with the layer of the cornea that the lesion occurred in and the presence of fluorescein (as seen on slit lamp analysis), the group generated 12 distinct characteristics.

The characteristics identified on SDAS-OCT were grouped by the causal microbes to try and identify patterns in infectious keratitis. For example, the localized small stromal cystic spaces interpreted as localised stromal necrosis and full-thickness large stromal cystic spaces as diffuse stromal necrosis were found to be only associated with fungal infections due to *Aspergillus* species. On the other hand, diffuse stromal thinning with an epithelial defect and positive fluorescein staining was only found with *Staphylococcus aureus* infections. Therefore, SD AS-OCT has the potential to assist in the identification of the causal microbe in cases of IK. This study was limited due to a small sample size, such that a larger study is needed to determine the sensitivity and specificity of such features for each type of keratitis [7,35].

In a case series of four eyes, Yamazaki et al. proposed that SD-OCT (Spectralis AS-OCT, Heidelberg, Germany) could be used to identify Acanthamoeba in vivo using corneal imaging [7,36]. This was completed by correlating highly reflective bands or oblique lines in the corneal stroma with the areas that corresponded to the infiltrate from an established confocal microscopy method, as well as to slit lamp images. A unique OCT pattern was found for Acanthamoeba that needs to be validated in future studies [7,36].

#### 3.1.2. Swept Source

Swept-source OCT (SS AS-OCT) scans faster than SD AS-OCT with speeds of up to 200,000 A-scans/s with modern commercial devices and millions of A-scans/s with laboratory devices. This type of tomography utilises a laser that rapidly sweeps through frequencies across a broad spectrum opposite to SD AS-OCT, which utilises a broad-bandwidth light source [33,38]. SS AS-OCT allows high scan speeds (shorter scan speeds and higher scan density), less depth-dependent signal-to-noise ratio and resolution drop-off, and improved scan quality (less eye movement). Most SS AS-OCT devices also utilise a centre wavelength of approximately 1050 nm (SD AS-OCT uses a centre wavelength of approximately 850 nm), which allows for greater axial depth imaging and better visualisation of deeper ocular structures, such as the choroid and the lamina cribrosa (LC) (Table 2) [33,39,40].

A study from Sydney, Australia, with 68 patients, investigated a link between SS AS-OCT and clinical outcomes by using a severity index scoring tool. Patients presenting with infectious keratitis were assessed with an SS AS-OCT on the day of presentation and after 6 days. Clinical signs of the infection, such as corneal thickness, corneal infiltrates, corneal ulceration and epithelial defects, were identified and assessed (Figure 4). These clinical features were utilised to develop a severity score based on a point system (Table 3). Clinical features of infectious keratitis were identified on OCT analysis in all cases on day zero and 48 (71%) cases on day six (Figure 5). The most common findings were epithelial defect (100%) and stromal oedema noted as corneal thickening. Corneal thickness changes were found in 57 (84%) cases on day zero. Of these, 25 (37%) had a change between 5–10% of the corneal thickness, 18 (32%) between 10–30%, 9 (16%) between 30–50%, and 5 (9%) over 50%. On day zero, corneal thickness changes were found in 89% of *Pseudomonas aeruginosa*, 26% of *Staphylococcus* spp. and 100% of *Streptococcus pneumoniae* cases [7].

Nineteen of 68 (28%) patients in this study presented complications from infectious keratitis. Twelve patients required tarsorrhaphy or corneal glueing; six, deep anterior lamellar keratoplasty (DALK); and one, vitrectomy. The average score of the surgical patients was 19. The patients who needed surgical interventions had a significantly higher score than those who resolved without intervention (*p* = 0.042). There was no statistical association between a single feature of AS-OCT and a surgical outcome. There was a significant correlation between patients whose scores on day six were the same or higher than day zero and the requirement of surgery (*p* = 0.003). Patients with improvement in severity scores from day zero to day six were more likely to not need surgical intervention (*p* = 0.027). This study showed that apparent features of AS-OCT analysis correlated with clinical outcomes and the need for surgical intervention [7]. Further work is needed to validate these findings for clinical use.

## 4. In Vivo Confocal Microscopy

In vivo confocal microscopy (IVCM) provides a high-resolution, in vivo assessment of corneal structures and pathologies at a cellular and subcellular level [19]. IVCM provides corneal images with 1 µm resolution of the three cornea layers, nerves and cells and is sufficient to produce images larger than a few micrometres of filamentous fungi or Acanthamoeba cysts [4,14,15,41]. A third-generation laser scanning confocal microscope Heidelberg Retinal Tomograph (HRT3) in conjunction with the Rostock Cornea Module (RCM) (Heidelberg Engineering, Germany) utilised 670 nm red wavelength and produced high-resolution images with lateral resolution close to 1 µm, axial resolution of 7.6 µm and 400× magnification (Table 2). This advance permitted the identification of yeasts, which first-generation confocal microscopes could not resolve [14,15,19,42,43].

IVCM has mainly been used in the evaluation of fungal and Acanthamoeba keratitis (AK) due to its axial limitation of 5–7 µm, which is not sufficient to detect bacteria (less than 5 µm) and viruses (in nanometres) (Table 2) [4,19,43]. IVCM has been a great ally to microbiological diagnostic tests as it can identify these organisms rapidly, overcoming the test’s variable positive rate of between 40–99% and a turnaround time of up to 2 weeks [19]. Therefore, IVCM is an imaging diagnostic test that is valuable in guiding initial therapy.

IVCM sensitivity ranges between 66.7% and 94%, and specificity between 78% and 100% in fungal keratitis [4,14,43,44,45,46]. *Aspergillus* spp. and *Fusarium* spp. are the main causal organisms of fungal keratitis [4,43]. With IVCM, *Aspergillus* spp. are identified as 5–10 µm in diameter and have septate hyphae with 45-degree angle dichotomous branches. On the other hand, *Fusarium* spp. typically branch at an angle of 90 degrees. In comparison, basal corneal epithelial nerves have more regular branching than hyper-reflective elements, and stromal nerves’ are between 25–50 µm in diameter versus *Aspergillus spp.* and *Fusarium* spp. diameter of 200–400 µm in length [19,43]. Yeast-like fungi (*Candida spp*.) can also cause keratitis with a predilection for the immunosuppressed (Figure 6); with IVCM, they appear as elongated, hyper-reflective particles resembling pseudohyphae of 10–40 µm in length and 5–10 µm in width (Table 2) [19,43].

IVCM is a key diagnostic test in AK with an overall sensitivity of 80–100% and specificity of 84–100% [19,44,45,46,47,48]. *Acanthamoeba* spp. can present as cysts or trophozoites. Cysts (dormant form) appear as hyper-reflective, spherical and well-defined double-wall structures of ~15–30 µm in diameter in the epithelium or stroma. Trophozoites (active form) appear as hyper-reflective structures of 25–40 µm, which are difficult to discriminate from leukocytes and keratocyte nuclei (Table 2) [4,19,49]. *Acanthamoeba* spp. can also present as bright spots, signet rings and perineural infiltrates (Table 2). Perineural infiltrates are a pathognomonic characteristic of AK, which appear as reflective patchy lesions with surrounding hyper-reflective spindle-shaped materials [19] (Figure 7).

The advantages of IVCM include ‘non-invasiveness’: the ability to rapidly identify in real time the causal organism and to determine the depth of the infection. This can guide the antimicrobial therapy and assist in monitoring the infection. Early identification and treatment of AK have been associated with improved prognosis [4,16]. Imaging by IVCM also facilitates longitudinal exams in the same patient, which may be of use in determining resolution and provides quantitative analysis of all cornea layers, nerves and cells to assess severity. The disadvantages of IVCM include the need for an experienced technician, patient cooperation, unsuitability for diagnosing bacteria and viruses due to axial limitation of 5–7 µm, high costs and the presence of motion artefacts. In addition, dense corneal infiltrates and/or scarring can affect proper tissue penetration and visualisation (Table 2) [4,14,15,41,43].

In clinical practice, typically, IVCM is performed in cases of progressive keratitis, suspicion of AK or FK, negative culture results, and in deep infections or interface infectious keratitis following corneal surgeries due to the limited access for conventional microbiological tests (Table 2) [4,19].

## 5. Artificial Intelligence—Deep Learning Methods

### 5.1. Background

The applications of artificial intelligence in health care are now a reality due to the advancement of computational power, refinement of learning algorithms and architectures, availability of big data and easy accessibility to deep neural networks by the public [50,51,52]. Deep learning algorithms mostly use multimedia data (images, videos and sounds) and involve the application of large-scale neural networks, such as artificial neural networks (ANN), convolutional neural networks (CNN) and recurrent neural networks (RNN) [51]. The advantage of deep CNNs is that they enable learning from data without human knowledge and the capability of processing large training data with high dimensionality [53]. A CNN model contains multiple convolutional layers, pooling layers and activation units, which are trained using model images by minimising a pre-defined loss function. A convolutional layer applies a number of filters to the input image calculated from the previous layer. This results in enhanced features at certain locations in the image. The weights in these filters are learned during the training process. A pooling layer is a dimensionality reduction operator that down-samples the input image obtained from the previous layer. Average and maximum are the most popular pooling operators, which calculate the average and max value of a local region during the image down-sampling process, respectively [53].

Some common terminologies used in AI studies are shown in Table 4 [53]. The sensitivity (*y*-axis) and specificity (*x*-axis) rates are used to illustrate the receiver operating characteristic (ROC) curve and the area under the ROC curve (AUROC), which are used to determine the performance of a model at all thresholds. An AUROC of 1 is a ‘perfect’ classifier, AUROC between 0.5–1 is a real-world classifier, and an AUROC of 0.5 is a ‘poor’ classifier, which is not better than a random guess [53].

The DL CNN model training needs to consider several aspects. For instance, the quality of the training dataset is essential to the performance of the DL CNN model. Image annotation refers to the process of labelling images of a dataset to train the DL models. The image annotation is given by the clinicians and usually includes pixel-level annotation, image-level annotation or both. Further, the model may suffer from ‘over-fitting’; that is, it cannot be generalised well to new test data due to many model parameters and relatively small training examples. A validation dataset is generally used to determine the training termination point to avoid model over-fitting. Drop-out, data augmentation, and transfer learning have been used to improve the generalisability of a trained model. An independent external test set is used to evaluate the trained DL model for assessing the generalisability of the method. A lower performance normally occurs when testing on an independent test set, which is mainly due to model overfitting to the training dataset or a data distribution discrepancy between the training and testing datasets. Figure 8 illustrates an example of a framework of several DL models to diagnose infectious keratitis based on Kuo et al. [54]

In 2016, the use of artificial intelligence (AI) with deep learning (DL) in ophthalmology initially focused on posterior segment diseases such as diabetic retinopathy and age-related macular degeneration, but its application in diseases of the cornea, cataract and anterior chamber structures has surged in the last years [50]. Corneal AI research has focused on diseases that require corneal imaging for determining appropriate management and has utilised slit lamp photography, corneal topography and anterior segment optical coherence tomography [50]. For infectious keratitis, the use of DL with CNNs has been shown to be a potentially more accessible diagnostic method via image recognition [4,54,55]. Many studies have evaluated DL methods for diagnosing IK using images taken with a handheld camera, a camera mounted on a slit lamp or confocal microscopy [51,55,56,57,58,59,60,61,62,63,64]. Several extremely efficient DL algorithms include RestNet-152 [65], DenseNet-169 [66], Mobile-Net V2 [67] and VGG-19_BN [68]. Recent studies have used an ensemble approach; this uses many learning algorithms to perform the same classification task, yielding better validity and improved generalisation performance [61,69]. The CNN-based ensemble approach takes elements of the different CNN algorithms to build the final model. Some studies have reported that the ensemble DL model could better classify the stages of diabetic retinopathy and glaucoma using fundus photos than a single CNN model [61].

### 5.2. Deep Learning Models in Infectious Keratitis

Table 5 shows several studies that have used DL models to diagnose IK. Li et al. developed a DL system to classify corneal images in keratitis, other corneal abnormalities and normal cornea. The authors used three DL algorithms to train an internal image dataset and had three external and smartphone datasets to externally evaluate the DL system. In terms of the smartphone dataset, the DenseNet121 algorithm elicited the best performance in classifying keratitis, other corneal abnormalities and normal cornea with an AUROC of 0.967 (95% CI, 0.955–0.977), a sensitivity of 91.9% (95% CI, 89.4–94.4) and a specificity of 96.9% (95% CI, 95.6–98.2) in detecting keratitis [70]. To differentiate the types of infectious keratitis, Redd et al. used images from handheld cameras to train five CNNs to differentiate FK from BK and compare their performance against human experience. The best-performing CNN was MobileNet, with an AUROC of 0.86. The CNNs group achieved a statistically significant higher AUROC (0.84) than the experts (0.76, *p* < 0.01). CNNs elicited higher accuracy for FK (81%) versus BK (75%) compared to the experts who showed more accuracy for BK (88%) versus FK (56%) [55]. Wang et al. investigated the potential of DL in classifying IK using slit lamp images and smartphone photographs. They also studied whether any information from the sclera, eyelashes and lids could improve the diagnosis of keratitis. The slit lamp images (global image) were pre-processed, excluding the irrelevant background (regional image). For the smartphone photographs, a small patch was extracted to make it look similar to the global slit lamp image; three CNNs were assessed. The InceptionV3 showed a better performance with an AUROC of 0.9588 on global images, 0.9425 on regional images and 0.8529 for smartphone images, while the two ophthalmologists reached an AUROC of 0.8050 and 0.7333, respectively. The lower performance of the smartphone images could be due to multiple factors, including the size of the dataset and the significant variation in imaging conditions (e.g., ambient light, camera brand and focus) [56].

Hung et al. used slit lamp images to train eight CNNs to identify BK and FK. The diagnostic accuracy for BK ranged from 79.6% to 95.9%, and for FK, 26.3% to 65.8%, respectively. The best-performing model was DenseNet161, with an AUROC of 0.85 for both types of keratitis. The diagnostic accuracy for BK was 87.3%, and for FK, 65.8%. [57]. Kuo et al. aimed to assess the performance of eight CNNs (four EfficientNet and four non-EfficientNet CNNs) in diagnosing BK using slit lamp images. All non-EfficientNet and EfficientNet models had no significant difference in diagnostic accuracy, ranging from 68.8% to 71.7% and an AUROC from 73.4% to 76.5% [54]. EfficientNet B3 had the best average AUROC with 74% sensitivity and 64% specificity. The diagnostic accuracy of these models (69% to 72%) was comparable to the ophthalmologists (66% to 75%) [54]. Ghosh et al. created a model called DeepKeratitis on top of three pre-trained CNNs to classify FK and BK. The CNN model VVG19 showed the highest performance with an F1 score of 0.78, precision of 0.88 and sensitivity of 0.70. Applying the ensemble learning model achieved higher performance with an F1 score of 0.83, precision of 0.91 and sensitivity of 0.77. The F1 score is a harmonic mean of precision and recall and measures a test performance, a value close to 1.0 indicates high precision and recall. The ensemble model achieved the highest AUPRC of 0.904 [62].

Hu et al. proposed a DL system with slit lamp images to automatically screen and diagnose IK (BK, FK and viral keratitis (VK)). Six CNNs were trained. The EffecientNetV2-M showed the best performance with 0.735 accuracy, 0.68 sensitivity and 0.904 specificity, which was also superior to two ophthalmologists (accuracy of 0.661 and 0.685). The overall AUROC of the EffecientNetV2-M was 0.85, with 1.00 for normal cornea, 0.87 for VK, 0.87 for FK and 0.64 for BK [71]. Kuo et al., in 2022, explored eight single and four ensemble DL models to diagnose BK caused by *Pseudomonas aeruginosa.* The EfficientNet B2 model reported the highest accuracy (71.2%) of the eight single-DL models, while the best ensemble 4-DL model showed the highest accuracy (72.1%) with 81% sensitivity and 51.5% specificity among the ensemble models. EfficientNetB3 had the highest specificity of 68.2%. There was no statistical difference in AUROC and diagnostic accuracy among these single-DL models and among the four best ensemble models [61]. Natarajan et al. explored the application of three DL algorithms to diagnose herpes simplex virus (HSV) stromal with ulceration keratitis using slit lamp images. DenseNet had the best performance with 72% accuracy. The AUROC was 0.73 with a sensitivity of 69.6% and specificity of 76.5% [63].

Koyama et al. developed a hybrid DL algorithm to determine the causal organism of IK by analysing slit lamp images. Facial recognition techniques were also used as they accommodate different angles, different levels of lighting and different degrees of resolution. ResNet-50 and InceptionResNetV2 were used. The final model was built based on InceptionResNetV2 using 4306 images consisting of 3994 clinical and 312 web images. This algorithm had a high overall accuracy of diagnosis: accuracy/AUROC for Acanthamoeba was 97.9%/0.995, bacteria was 90.7%/0.963, fungi was 95.0%/0.975, and HSV was 92.3%/0.946 [58]. Similarly, Zhang et al. aimed to develop a DL diagnostic model to early differentiate bacterial, fungal, viral and Acanthamoeba keratitis. KeratitisNet, the combination of ResNext101_32 × 16 d and DenseNet169 had the highest accuracy, 77.08%. The diagnostic accuracy/AUROC was 70.27%/0.86, 77.71%/0.91, 83.81%/0.96 and 79.31%/0.98 for BK, FK, AK and HSK, respectively. KeratitisNet mostly misinterpreted BK and FK images, with 20% of BK cases mispredicted into FK and 16% of FK cases into BK. The accuracy of the model was significantly higher than an ophthalmologist’s clinical diagnosis (*p* < 0.001) [59].

Several studies have also investigated the performance of DL CNNs methods in diagnosing FK using IVCM images [50,51]. Liu et al. proposed a novel CNN model for automatically diagnosing FK using data augmentation and image fusion. The accuracy of conventional AlexNet and VGGNet were 99.35% and 99.14%. This CNN model perfectly balanced diagnostic performance and computational complexity, improving real-time performance in diagnosing FK [72]. Lv et al. developed an intelligent system based on the DL CNN (ResNet) model to automatically diagnose FK using IVCM images. The AUROC of the system to detect hyphae was 0.9875 with an accuracy of 0.9364, sensitivity of 0.8256 and specificity of 0.9889 [73].

### 5.3. Future Perspectives

Up to now, the majority of studies investigating the use of AI in ophthalmology have focused on disease screening and diagnosis using existing clinical data and images based on machine learning and CNNs in conditions such as AMD, diabetic retinopathy, glaucoma and cataract [74]. For infectious keratitis diagnosis, the generation of synthetic data using generative adversarial networks (GAN) may be a new method to train AI models without the need for thousands of images from real cases used in CNNs. In the case of less common conditions like fungal or Acanthamoeba keratitis, a GAN could be utilised as a low-shot learning method via data augmentation, meaning that conventional DL models could learn less common conditions using a low number of images [75,76]. The low-shot learning technique has been used in detecting and classifying retinal diseases [77,78] and in conjunctival melanoma [75].

Another AI technology that generates synthetic data is natural language processing (NLP) models, such as ChatGPT, developed by OpenAI (San Francisco, CA, USA) [74,79]. ChatGPT utilises DL methods to generate logical text based on the user’s ‘prompt’ in layman’s terms [80]. ChatGPT was not conceived for specific tasks, such as reading images or assessing medical notes; however, OpenAI has investigated the potential use of ChatGPT in healthcare and medical applications and research. Some applications include medical note-taking and medical consultations. The medical knowledge embedded in ChatGPT may be utilised in tasks such as medical consultation, diagnosis, and education with variable accuracy [81]. For example, Delsoz et al. entered corneal medical cases (including infectious keratitis) on ChatGPT 4.0 and 3.5 to obtain a medical diagnosis, which was compared with the results from three corneal specialists. The provisional diagnosis accuracy was 85% (17 of 20 cases) for ChatGPT-4.0 and 65% for ChatGPT-3.5 versus 100% (specialist 1) and 90% (specialists 2 and 3, each) [79]. As a result, ChatGPT may be utilised to analyse clinical data along with DL models (CNNs or GAN) to diagnose and differentiate infectious keratitis.

### 5.4. Limitations

Despite the great advancement of DL with CNNs in the diagnosis of conditions using images, it is still challenging to achieve a satisfactory diagnostic performance in IK. Reasons for this include the large intra-class variance (difficulty in capturing the common characteristics of images in the same class), small inter-class difference (difficulty in discriminating the margin between different classes), non-standard image protocols increasing the difficulty of finding the common special features of the different types of IK [64]; difficulty in overfitting due to the lack of large scale dataset in IK [63,64], imbalance in the dataset to train models [57] and misclassification bias [62]. To overcome some of the limitations, Li et al. developed a novel CNN called Class Aware Attention Network to diagnose IK using slit lamp images. In this network, a class-aware classification module is first trained to learn class-related discriminative features using separate branches for each class. Next, the learned class-aware discriminative features are fed into the main branch and fused with other feature maps using two attention strategies to assist the final multi-class classification performance [64]. This method showed a higher accuracy (0.70) and specificity (0.89) compared to eight single CNNs. This innovative CNN can extract the fine features of keratitis lesions, which are not easy for clinicians to identify [64]. Further, there is a wide range of causal organisms in keratitis and varied clinical presentations with regional differences in both [4,58,82,83]. Improving image resolution, increasing the number of images for the training models, and optimising the parameters of the algorithms may enhance the accuracy of the models [51,72,73]. Future studies will be needed to validate the findings of CNN approaches to the diagnosis of IK in a range of global settings with a variety of devices (slit lamp microscopy and camera) [73].

## 6. Conclusions

Infectious keratitis is among the top five leading causes of blindness worldwide. Early diagnosis of the causal organism is crucial to guide management to avoid severe complications, such as vision impairment and blindness. Clinical examination under the slit lamp is essential for diagnosing the infection, and the culture of corneal scrapes is still the gold standard in the identification and isolation of the causal organism. Slit lamp photography was revolutionised by the invention of digital cameras and smartphones. Slit lamp image quality essentially depends on the digital camera or smartphone camera sensor’s resolution, the resolution of the slit lamp and the focal length of the smartphone camera system. An image of six megapixels is sufficient in clinical photography. Alternative diagnostic tools using imaging devices, such as OCT and IVCM, can track the progression of the infection and may identify the causal organism(s). The overall sensitivity and specificity of IVCM for fungal keratitis are 66–74% and 78–100%, and for Acanthamoeba keratitis, they are 80–100% and 84–100%.

Due to the advancement of AI technology, many study groups worldwide have evaluated the role of DL-CNN models to diagnose the different types of IK using either corneal photographs taken with smartphones or slit lamp cameras or confocal images with variable promising results. Nevertheless, there are some challenges to overcome to use this technology in health settings, including non-standard imaging protocols, difficulty in finding the common special features of the different types of IK, cost, accessibility and cost-effectiveness. More recent DL models, such as GAN, may help in overcoming these limitations by generating synthetic data to train the models. Moreover, natural language processing models, such as ChatGTP, may analyse clinical data. These results, along with GAN results, may assist in diagnosing IK and differentiating the types of IK. Further work is needed to examine and validate the clinical performance of these CNNs models in real-world healthcare settings with multiethnicity populations to increase the generalisability of the model. Finally, it would also be valuable to evaluate whether this technology assists in improving patient clinical outcomes via prospective clinical trials.

## 7. The Literature Search

We conducted a Pubmed and Ovid Medline search of the literature from 2010 to 2022. The terms used covered a wide spectrum of terms associated with corneal imaging in infectious keratitis, including cornea, corneal ulcer, microbial keratitis, infectious keratitis, bacterial keratitis, fungal keratitis, HSV keratitis, viral keratitis, herpetic keratitis, optical coherence tomography, deep learning, convolutional neural networks, slit lamp images, artificial intelligence, computer vision, machine learning and in vivo confocal microscopy. Reference lists from the recovered articles were also used to identify cases that may not have been included in our initial search.

## Figures and Tables

**Figure 1 diagnostics-13-03358-f001:**
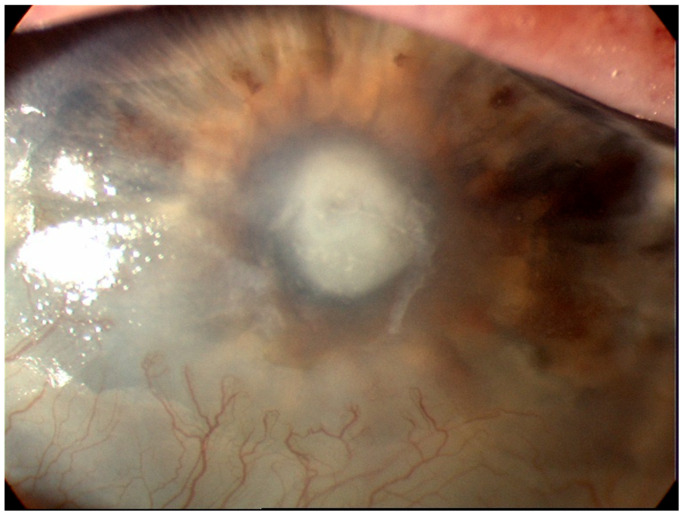
Corneal ulceration in bacterial keratitis. Note there is an infiltrate underlying an epithelial defect (Image courtesy of Prof. Stephanie Watson).

**Figure 2 diagnostics-13-03358-f002:**
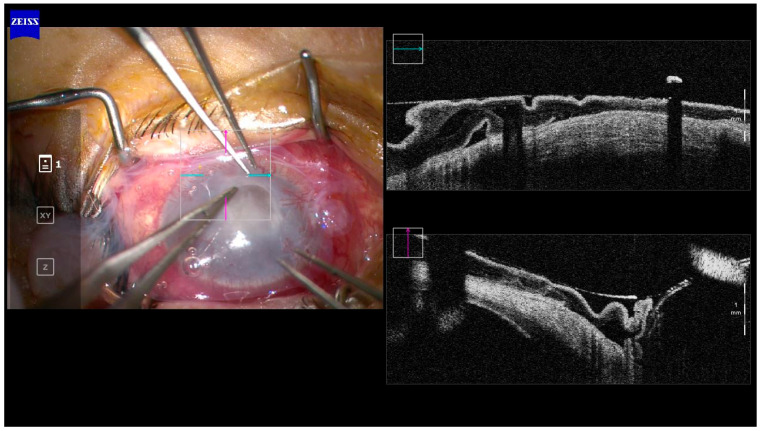
Intra-operative OCT in a case of amniotic membrane transplant for a persistent epithelial defect post-microbial keratitis. Corneal thinning can be seen in the area of the defect. (Image courtesy of Prof. Stephanie Watson).

**Figure 3 diagnostics-13-03358-f003:**
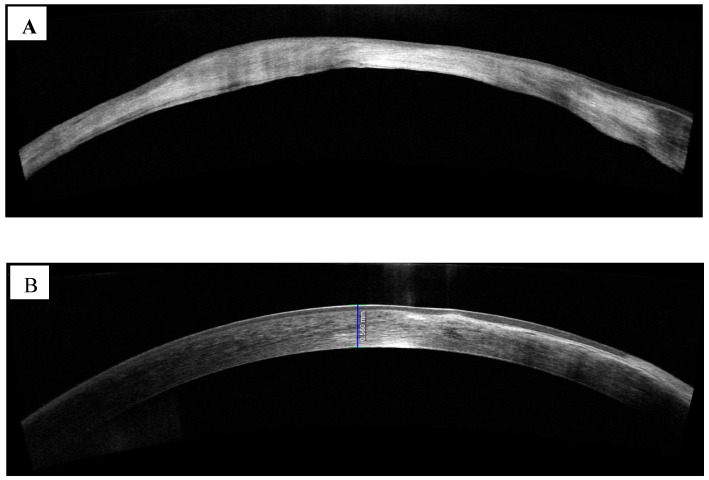
(**A**,**B**): SDAS-OCT images of the cornea showing (**A**) Corneal scarring and thinning in a patient following acanthamoeba keratitis and (**B**) corneal thickness measurement and epithelial layer imaging. (Image courtesy of Prof. Stephanie Watson).

**Figure 4 diagnostics-13-03358-f004:**
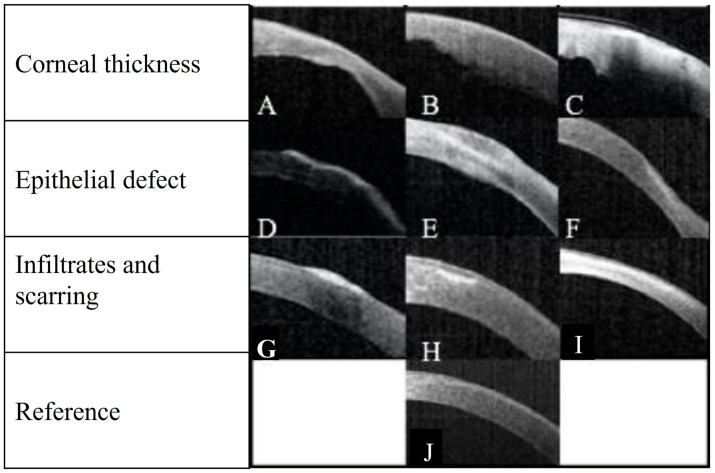
AS-OCT images demonstrating the characteristics of infectious keratitis. (**A**) Corneal thinning 30–50% with stromal oedema and epithelial defect. (**B**) Corneal thinning 5–10% with stromal oedema. (**C**) Stromal thinning 30–50%, and stromal oedema. (**D**) Long epithelial defect with an underlying hyper-reflective region that would progress to Descemetocele. (**E**) Epithelial defect with underlying scarring. (**F**) Epithelial defect with infiltrate. (**G**) Epithelial defect with infiltrate and hypo-reflective region. (**H**) hyperreflective infiltrate (**I**) Corneal thinning with scarring. (**J**) Example of reference in an unaffected eye [7].

**Figure 5 diagnostics-13-03358-f005:**
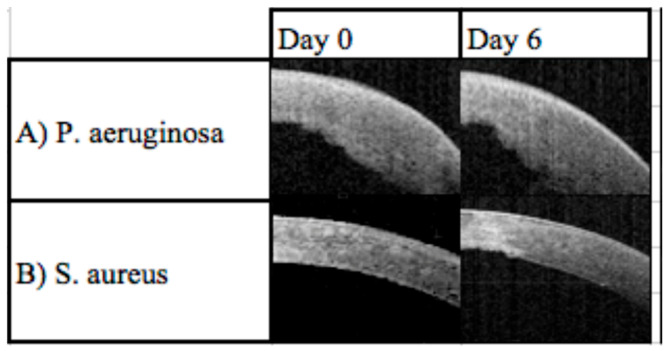
Example of serial features. (**A**) *Pseudomonas aeruginosa* on day zero and day six with stromal oedema, epithelial defect and poor progression. The outcome, in this case, was surgery. Note the increased corneal thickness and opacity. (**B**) *Staphylococcus aureus* at day zero and day six with infiltrates and scar [7]; at day six, the white area indicating scarring and corneal thinning has resulted from the resolution of the oedema.

**Figure 6 diagnostics-13-03358-f006:**
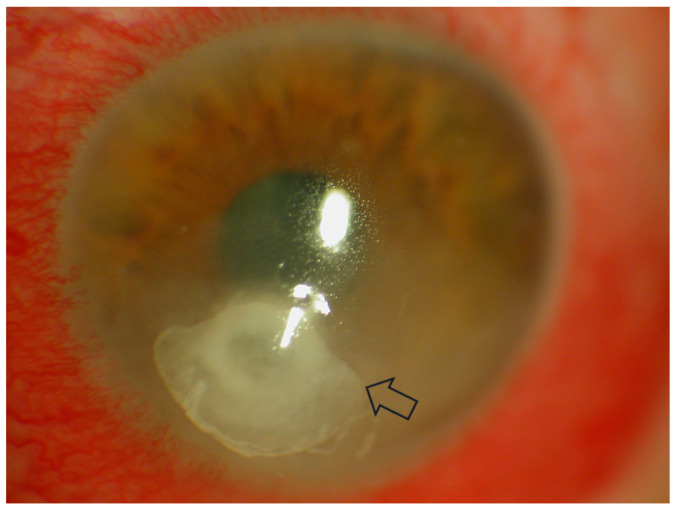
The clinical appearance of *Candida albicans* keratitis in a patient on immunosuppression. The arrow is pointing to the area of infiltrate. (Image courtesy of Prof. Stephanie Watson).

**Figure 7 diagnostics-13-03358-f007:**
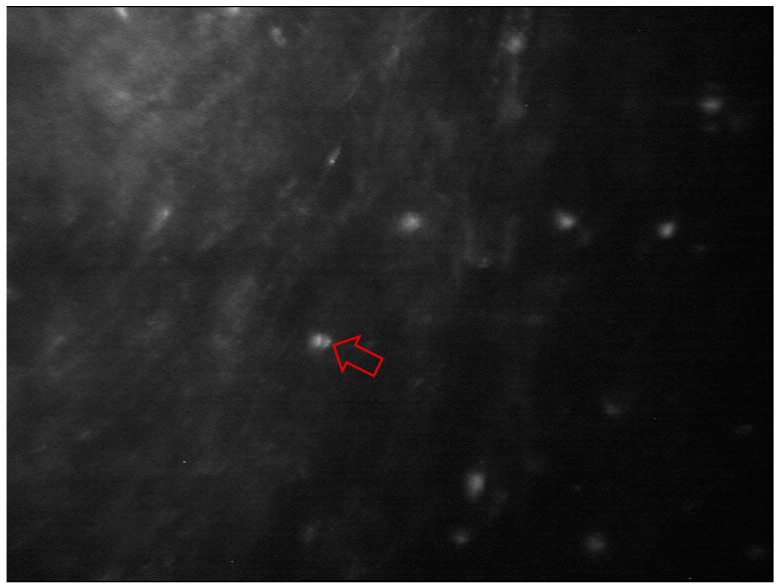
In vivo confocal microscopy of Acanthamoeba keratitis. Perineural infiltrates appear as a reflective patchy lesion with surrounding hyperreflective spindle-shaped materials (red arrow). Image courtesy of Prof. Stephanie Watson.

**Figure 8 diagnostics-13-03358-f008:**
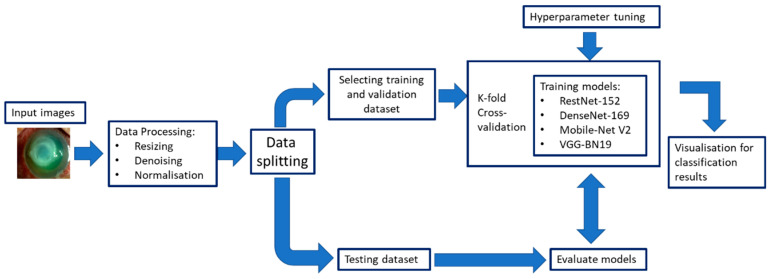
The framework of several deep-learning models to diagnose bacterial keratitis was inspired by the framework of Kuo et al. [54].

**Table 1 diagnostics-13-03358-t001:** Most common causal organisms and clinical signs of infectious keratitis. Key: AK = acanthamoeba keratitis, BK = bacterial keratitis, FK = fungal keratitis, HSK = herpes simplex keratitis.

	BK	HSK	FK	AK
Most causal organism	- *Staphylococci* spp., - *Pseudomonas aeruginosa* - *Streptococcus pneumoniae*.	Herpes simplex virus	- Filamentous (*Fusarium* spp., *Aspergillus* spp.) - Yeast (*Candida* spp.)	*Acanthamoeba* spp.
Clinical signs	- Corneal lesion - Corneal infiltrates - hypopyon	- Epithelial: dendritic or geographic ulcer - Stromal: Haze with or without ulcer, lipid keratopathy, stromal oedema, scarring, corneal thinning or vascularisation - Endothelial: Stromal oedema and keratic precipitates - Keratouveitis: Stromal oedema, keratic precipitates and anterior chamber cells	- Corneal ulcer with irregular feathery margins - Elevated borders or dry, rough texture - Satellite lesions - Descemet’s folds, hypopyon, - Ring infiltrate - Endothelial plaque - Anterior chamber cells - Keratic precipitates	- Corneal ulcer with ring infiltrates - Deep stromal infiltrates - Corneal perforation - Satellite lesions - Scleritis - Anterior uveitis with hypopyon

**Table 2 diagnostics-13-03358-t002:** Summary of available imaging diagnostic methods in infectious keratitis. Key: AK = Acanthamoeba keratitis, FK = fungal keratitis, IK = infectious keratitis, HRT = Heidelberg Retinal Tomograph, IVCM = In vivo confocal microscopy, SS-OCT = Swept source optical coherence tomography.

Imaging Method	Clinical Features	Resolution of Images	Clinical Indications	Sensitivity	Specificity	Advantages	Disadvantages
SS OCT	Corneal thickness change Epithelial defect Hypopyon Infiltrates Stromal thinning Cysts Fibrin deposits	Speed up to 200,000 A-scans/s (commercial devices) Millions of scans (laboratory devices) Centre wavelength of approximately 1050 nm				Allows high scan speeds (short scan speed and higher scan density) Improved scan quality Less depth-dependant signal-to-noise ratio and resolution drop-off Better resolution of deep ocular structures	
IVCM	• Filamentous FK: – *Aspergillus* spp. = 5–10 µm in diameter, septate hyphae with 45-degree angle dichotomous branches – *Fusarium* spp. = branches at 90-degree angle – Nerves = Basal corneal epithelial nerves with more branching than hyperreflective elements – Stromal nerves = 25–50 µm diameter versus filamentous fungi with 200–400 µm diameter • Yeast (*Candida*) FK: hyper-reflective particles resembling pseudohyphae of 10–40 µm in length and 5–10 µm in width. • AK: – Cysts = hyperreflective, spherical and well-defined double-wall structures of ~15–30 µm in diameter in the epithelium or stroma. – Trophozoites = hyper-reflective structures of 25–40 µm, difficult to discriminate from leukocytes and keratocyte nuclei. – Acanthamoeba can present as right spots, signet rings and perineural infiltrates.	HRT3 uses 670 nm red wavelength and high-resolution images with lateral resolution close to 1 µm, axial resolution of 7.6 µm and 400× magnification	Progressive keratitis. Suspicion of AK or FK Negative culture result Deep infection due to natural clinical course or FK or development of interface IK following corneal surgeries limiting access to conventional microbiological tests	FK= 66.7–94% AK = 80–100%	FK = 78–100% AK = 84–100%	Non-invasiveness Rapid, real-time Early identification of the organism for monitoring and guidance of the therapy. Determination of the depth of the infection.	• Axial limitation of 5–7 µm to detect bacteria and viruses. • Need for an experienced operator • Patient cooperation • Motion artefacts • High costs • Dense corneal infiltrates and/or scarring can affect the proper tissue penetration and visualisation

**Table 3 diagnostics-13-03358-t003:** Point system of the characteristics assessed using AS-OCT in cases of infectious keratitis [7]. Key: AS-OCT = anterior segment optical coherence tomography.

Characteristic on AS-OCT	Points
Corneal thickness change 5–10%	1
Corneal thickness change 10–30%	2
Corneal thickness change 30–50%	3
Corneal thickness change >50%	4
Epithelial defect 0.1–1 × 0.1–1	1
Epithelial defect 1–2 × 1–2	2
Epithelial defect 2–3 × 2–3	3
Epithelial defect >3x > 3	4
Hypopyon	1
Infiltrates 0.1 mm^–1^	1
Infiltrates >1	2
Stromal thinning	1
Cysts	1
Scarring	1
Fibrin deposits	1
Total	28

**Table 4 diagnostics-13-03358-t004:** (**A**) Definitions of common terminologies in disease screening; (**B**) Definitions of common terminologies in artificial intelligence studies [53].

(**A**)		
	Actual Outcome	
	Disease	No Disease
Predicted outcome	Disease	TP
	No Disease	FN
(**B**)		
Sensitivity	TP/(TP + FN)	Actual positive cases predicted correctly by the model
Specificity	TN/(TN + FP)	Actual negative cases predicted correctly by the model
PPV (Precision)	TP/(TP + FP)	Positively classified cases that were actually positive
NPV	TN/(TN + FN)	Negatively classified cases that were actually negative
Accuracy	(TP + TN)/(TP + TN + FP + FN)	Overall accuracy in predicting both positive and negative cases.

TP = True positive; FP = False positive; FN = False negative; TN = True negative; PPV = Positive predictive value; NPV = Negative predictive value.

**Table 5 diagnostics-13-03358-t005:** Summary table of deep learning models in infectious keratitis. Key: AUROC = area under the receiver operating curve, AK = acanthamoeba keratitis, AUPRC = area under the precision-recall curve, BK = bacteria keratitis, CNN = convolutional neural network, DN = DenseNet, EN = EfficientNet, FK = fungal keratitis, HSK = herpes simplex keratitis, IC = inception, IK = infectious keratitis, IVCM = in vivo confocal microscopy, MN = MobileNet, RN = ResNet, VK = viral keratitis. Bold figures mean the best algorithm performance.

Authors	Year	Study Population	Image Modality	Image Size	AI Algorithm	Outcome Measures	AUROC (95% CI)	AUPRC (95% CI)	F1 Score	Accuracy % (95% CI)	Sensitivity % (95% CI)	Specificity % (95% CI)
Li et al. [70]	2021	Healthy eyes	Smart phone (Huawei P30)	1030	DN121, IC-v3, RN50	Detect of keratitis, cornea with other abnormalities and normal cornea	- **DN121 = 0.967 (0.955–0.977)** - IC-v3 = 0.921 (0.902–0.936) - RNet50 = 0.953 (0.941–0.965)			- **DN121 = 94.9 (93.6–96.1)** - IC-v3= 85.9 (83.8–87.9) - RN50 = 90.5 (88.8–92.2)	- **DN121= 91.9 (89.4–94.4)** - IC-v3 = 70.6 (66.5–74.7) - RN50 = 84.6 (81.4–87.9)	- **DN121 = 96.9 (95.6–98.2)** - IC-v3 = 96.5 (95.1–97.8) - RN50 = 94.5 (92.8–96.2)
Redd et al. [55]	2022	Culture-proven BK or FK	Handheld Nikon D-series digital single-lens reflex camera	980	MNV2, DN201, RN152V2, VGG19, Xception	Different BK and FK	Multicentre set of 80 images: - **MNV2 = 0.86 (0.78–0.93)** - DN201 = 0.84 (0.76–0.92) - RN152V2 = 0.76 (0.67–0.85) - VGG19 = 0.74 (0.64–0.84) - Xception = 0.68 (0.57–0.78) - Humar graders = 0.76 (0.73–0.80) Single center set of 100 images: - MNV2 = 0.83 (0.74–0.92) - DN201 = 0.83 (0.74–0.92) - RN152V2 = 0.82 (0.72–0.91) - VGG19 = 0.75 (0.64–0.86) - Xception = 0.75 (0.64–0.86) - **Ensemble = 0.84 (0.76–0.92)**			CNN: - FK = 81 - BK = 75 Human grader ensemble: - BK = 88 - FK = 56		
Wang et al. [56]	2021	Normal eyes and with IK	Slit lamp-mounted camera and smart phone	- 5673 slit lamp photo- graphs - 400 smart- phone photo- graphs	IC-v3, RN50, DN121	Different normal cornea, BK, FK and VK	Slit lamp *Global images:* - IC-v3 = 0.9588 (94.28–97.48) - RN50 = 0.9517 (93.37–96.97) - **DN121 = 0.9605 (94.45–97.65)** Regional images: - **IC-v3 = 0.9425 (92.35–96.15)** - RN50 = 0.9369 (91.69–95.69) - DN121 = 0.9357 (91.57–95.57) *Ophthalmologists:* - 0.8050 (77.30–82.70) - 0.7333 (69.73–76.93) Smartphone *Images*: - **IC-v3 = 0.8529 (81.79–88.79)** - RN50 = 0.8188 (78.08–85.68) - DN121 = 0.7115 (66.75–75.55) *Ophthalmologists:* - 0.8467 (81.17–88.17) - 0.7100 (66.60–75.40)					
Hung et al. [57]	2021	Culture-proven IK	Slit lamp mounted camera	1330	RN50, RN101, DN121, DN 161, DN 169, DN201, IC-v3, ENB3	Identify BK and FK	- RN50 = 0.82 - RN101 = 0.77 - DN121 = 0.82 - **DN161 = 0.85** - DN 169 = 0.78 - DN201 = 0.80 - IC-v3 = 0.82 - ENB3 = 0.75			BK vs. FK: - RN50 = 95.9 vs. 26.3 - RN101 = 93.2 vs. 49.1 - DN121 = 85 vs. 59.6 - DN161 = 87.3 vs. 65.8 - DN169 = 79.6 vs. 63.2 - DN201 = 89.1 vs. 61.6 - IC-v3 = 89.1 vs. 61.6 - ENB3 = 85.2 vs. 58.1	- RN50 = 26.3 (15.5–39.7) - RN101 = 49.1 (35.6–62.7) - DN121 = 59.7 (45.8–72.4) - **DN161 = 65.8 (41.5–65.8)** - DN169 = 63.2 (49.32–75.6) - DN201 = 56.1 (42.4–69.3) - IC-v3 = 61.6 (50.5–71.9) - ENB3 = 58.1(47–68.7)	- **RN50 =** **95.9 (91.3–98.5)** - RN101 = 93.2 (87.9–96.7) - DN121 = 85 (78.2–90.4) - DN161 = 87.3 (86–95.3) - DN169 = 79.6 (72.2–85.8) - DN201 = 87.8 (81.3–92.6) - IC-v3 = 89.1 (82.3–93.9) - ENB3 = 85.2 (77.8–90.8)
Kuo et al. [54]	2021	Culture-proven IK or three specialists have a consensus impression of one type of IK	Slit lamp mounted camera	1512	RN50, RNXt50, SE-RN50, DN121, ENB0, ENB1, ENB2, ENB3,	Diagnose BK	- RN50 = 0.75 - RNXt50 = 0.74 - SE-RN50 = 0.75 - DN121 = 0.75 - ENB0 = 0.73 - ENB1 = 0.75 - ENB2 = 0.74 - ENB3 = 0.76			- RN50 = 69.3 (63.9–74.5) - RNXt50 = 69.8 (62.8–76.8) - SE-**RN50 = 71.7 (68.3–75)** - DN121 = 70.4 (67.3–73.4) - ENB0 = 68.8 (63.3–74.2) - ENB1 = 70.3 (65.8–74.8) - ENB2 = 69.6 (65.9–73.3) - ENB3 = 70.3 (64.6–75.9)	- RN50 = 81.1 (74.2–87.8) - RNXt50 = 79.1 (71.8–86.3) - SE-**RN50 = 82.4 (74.4–90.2)** - DN121 = 81.2 (74.2–88) - ENB0 = 74.4 (68.4–80.2) - ENB1 = 74.2 (66.7–81.5) - ENB2 = 73.5 (69.3–77.7) - ENB3 = 74.1 (64.6–83.5)	- RN50 = 50.4 (41.7–59.1) - RNXt50 = 55.1 (44.1–65.9) - SE-RN50 = 54.7 (47–62.4) - DN121 = 53.2 (46.8–59.4) - ENB0 = 59.9 (53.2–66.4) - ENB1 = 64.2 (59–69.3) - ENB2 = 63.5 (56.7–70.2) - **ENB3 = 64.3 (58.1–70.5)**
Ghosh et al. [62]	2022	Culture-proven IK	Slit lamp-mounted camera	2167	Deep-Keratitis model: VGG19, RN50, DN121,	Classify BK and FK		- VGG19 = 0.862 - RN50 = 0.59 - DN121= 0.728 - **Ensem-ble = 0.904**	- VGG19 = 0.78 (0.72–0.84) - RN50 = 0.68 (0.61–0.75) - DN121 = 0.71 (0.64–0.78) - **Ensem-ble = 0.83 (0.77–0.89)**		- VGG19 = 0.70 (0.63–0.77) - **RN50 = 0.85 (0.80–0.90)** - **DN121 = 0.85 (0.80–0.90)** - Ensemble = 0.77 (0.81–0.83)	
Hu et al. [71]	2023	Culture-proven IK	Slit lamp-mounted camera	2757	VGG16, RN34, IC-v4, DN121, Vit-Base, ENV2-M	Different BK, FK and VK	- VGG16 = 0.83 - RN34 = 0.82 - **IC-v4 = 0.86** - DN121 = 0.81 - Vit-Base = 0.82 - ENV2-M = 0.85			- VGG16 = 70.8 - RN34 = 63.5 - IC-v4 = 71.6 - DN121 = 63.7 - Vit-Base = 69.7 - **ENV2-M = 73.5**	- VGG16 = 0.583 - RN34 = 0.554 - IC-v4 = 0.640 - DN121 = 0.637 - Vit-Base = 0.598 - **ENV2-M = 0.68**	- VGG16 = 0.89 - RN34 = 0.861 - IC-v4 = 0.897 - DN121 = 0.875 - Vit-Base = 0.888 - **ENV2-M =** **0.904**
Kuo et al. [61]	2022	Culture-proven BK	Slit lamp mounted camera	929	RN50, RNXt50, SE-RN50, DN121, ENB0, ENB1, ENB2, ENB3,	Diagnose *Pseudomonas aeruginosa* keratitis				RN50 = 70.2 (64.4–75.9) RNXt50 = 69.8 (65.8–73.7) SE-RN50 = 70 (66.2–73.7) DN121 = 70.9 (65.7–76.1) ENB0 = 67 (60.3–73.6) ENB1 = 68.5 (64.3 -72.6) ENB2 = 71.2 (68.5–73.8) ENB3 = 68.6 (64.6–72.5)	RN50 = 80.4 (73.5–87.3) RNXt50 = 81.2 (74.6–87.9) SE-RN50 = 82.4 (68.7–96.1) DN121 = 82.5 (78.5–86.6) ENB0 = 66.5 (55.7–77.4) ENB1 = 74.6 (64.4 -84.8) ENB2 = 81.1 (76.3–85.8) ENB3 = 68.8 (62–75.5)	RN50 = 49.9 (39–60.7) RNXt50 = 46.9 (32.2–61.7) SE-RN50 = 45.3 (20.54–70.1) DN121 = 47.9 (37–58.8) ENB0 = 67.9 (62.2–73.5) ENB1 = 56.3 (47–65.6) ENB2 = 51.5 (47.1–55.8) ENB3 = 68.2 (65–71.3)
Natarajan et al. [63]	2022	Active HSV stromal necrotising keratitis PCR-proven and culture-proven nonviral keratitis	Slit lamp-mounted camera	307	DN201, RN-50, Google-Net	Diagnose HSK stromal necrotising	DN201 = 0.73 (0.568–0.892)			**DN201 = 72** RN-50 = 60	DN201 = 69.6	DN201 = 76.5
Koyama et al. [58]	2021	Culture-proven IK	Slit lamp mounted camera	4306	RN-50, IC, RN-v2	Determine the type of IK	IC+RN-v2: - **AK = 0.995 (0.991–0.998)** - BK = 0.963 (0.952–0.973) - FK = 0.975 (0.964–0.984) - HSK = 0.946 (0.926–0.964)			IC+RN-v2: - **AK = 97.9** - BK = 90.7 - FK = 95 - HSK = 92.3	IC+RNV2: - AK = 85 - **BK = 96** - FK = 88 - HSK = 62	
Zhang et al. [59]	2022	- Culture proven BK, IK, AK - Three corneal specialists diagnosed HSK	Slit lamp mounted camera	4830	RN18, RN50, DN121, DN169, EN-b0, EN-b5, EN-b7, RestNext 101_32 × 8 d, ResNext101_32 × 16 d	Differentiate BK, FK, HSK, AK	- BK = 0.86 - FK = 0.91 - AK = 0.96 - **HSK = 0.98**			- KeratitisNet (ResNext101_32 × 16 d+ DN169) = 77.08 BK = 70.27, FK = 77.71, **AK = 83.81**, HSK = 79.31		
Liu et al. [72]	2020	Healthy eyes and FK	IVCM	1213	AlexNet VGGNet	Detect FK				AlexNet = 99.35 VGGNet = 99.14	99.9	100
Lv et al. [73]	2020	Culture-proven FK	IVCM	2088	RN	Detect FK	0.9769			0.9364	0.8256	0.9889

## Data Availability

Not applicable.

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
