# Peer review of "Updates in Diagnostic Imaging for Infectious Keratitis: A Review"

_diagnostics, 2023, doi:10.3390/diagnostics13213358_

Round 1

Reviewer 1 Report

This article tried to review recent developments of diagnostic techniques (including AI) for keratitis. It is interesting issue but there are several critical concerns about the manuscript.

1.     The most important diagnostic tool is slit lamp (or external ocular image). The “external ocular image” chapter shall be added between introduction and OCT. Please see “Preventing corneal blindness caused by keratitis using artificial intelligence, nature communications, 2022”.

2.     Figure 1 does not appear to be acute bacterial keratitis. Is it neurotrophic keratitis?

3.     Anterior segment OCT is only an auxiliary tool to help diagnose keratitis. Clearly state why the authors want to take a multi-modal approach.

4.     Figure 2 has nothing to do with the diagnosis of keratitis.

5.     Many descriptions of OCT are concentrated on the device for poster segment. They need to be deleted. Clearly describe the type of as-oct (anterion, casia2, etc.).

6.     Figure 6 requires a corresponding confocal image.

7.     “Artificial intelligence - Deep learning methods” ***This chapter should review which previous studies have been conducted by image domain, and which corneal inflammation was diagnosed with what accuracy. Add a review summary table.

8.     Since it's a review article, clearly state what keyword you searched in which DB.

9.     The development of AI related to the image of the external eye continues. A lot of literature is missing from the current review and needs to be added.

10.  It is also necessary to discuss the direction of AI development in the future. In particular, in the case of keratitis, AI should be produced using a small number of data. This approach has been introduced, for example, “Adopting low-shot deep learning for the detection of conjunctival melanoma using ocular surface images” and “Deep Sequential Feature Learning in Clinical Image Classification of Infectious Keratitis”.

11.  Early diagnosis and differential diagnosis of keratitis (bacterial, fungal, viral, or other inflammation) using AI are important problems. Emphasize this issue in introduction and conclusion.

12.  The authors should predict more about the future diagnosis of keratitis with the development of generative artificial intelligence and chatgpt. Please refer the following articles: “Artificial intelligence-based ChatGPT chatbot responses for patient and parent questions on vernal keratoconjunctivitis” and “New era after ChatGPT in ophthalmology: advances from data-based decision support to patient-centered generative artificial Intelligence”.

Author Response

Reviewer 1

This article tried to review recent developments of diagnostic techniques (including AI) for keratitis. It is interesting issue but there are several critical concerns about the manuscript.

  1. The most important diagnostic tool is slit lamp (or external ocular image). The “external ocular image” chapter shall be added between introduction and OCT. Please see “Preventing corneal blindness caused by keratitis using artificial intelligence, nature communications, 2022”.

Thanks for the comment.

We agreed that the slit lamp is an essential tool in the diagnoses of infectious keratitis. In section 4, we commented on a study by Redd et al. where they used images taken with handheld cameras to train the Deep learning models. We have added a comment based on the paper you recommended on page 25:

Li et al. developed a DL system to classify automatically corneal images in keratitis, other corneal abnormalities, and normal cornea. The authors used three DL algorithms to train an internal image dataset and had other 3 external and a smartphone dataset to externally evaluate the DL system. In terms of the smartphone dataset, the DenseNet121 algorithm elicited the best performance in classifying keratitis, corneal with other abnormalities and normal cornea with an AUC of 0.967 (95% CI, 0.955-0.977), a sensitivity of 91.9% (95% CI, 89.4–94.4), and a specificity of 96.9% (95% CI, 95.6–98.2) in detecting keratitis (1).

  1. Figure 1 does not appear to be acute bacterial keratitis. Is it neurotrophic keratitis?

Thank you for the comment. The authors can confirm this is a case of bacterial keratitis; there is an infiltrate underlying an epithelial defect consistent with bacterial keratitis.

  1. Anterior segment OCT is only an auxiliary tool to help diagnose keratitis. Clearly state why the authors want to take a multi-modal approach.

The following sentence has been added on page 9 for clarity:

For bacterial keratitis, AS-OCT is currently an auxiliary tool to assist diagnosis.

  1. Figure 2 has nothing to do with the diagnosis of keratitis.

Thank you for the comment. Figure 2 is demonstrating the ability of OCT to assist with determining corneal thinning a sign of microbial keratitis. This has been added to the text and Figure legend.

  1. Many descriptions of OCT are concentrated on the device for poster segment. They need to be deleted. Clearly describe the type of as-oct (anterion, casia2, etc.).

Thanks for the comment. This was amended.

  1. Figure 6 requires a corresponding confocal image.

Thank you. The Figure legend has been edited to correspond more closely to the figure.

  1. “Artificial intelligence - Deep learning methods” ***This chapter should review which previous studies have been conducted by image domain, and which corneal inflammation was diagnosed with what accuracy. Add a review summary table.

Thanks for the comment. Table 3 was added as summary table of the DL studies. 

  1. Since it's a review article, clearly state what keyword you searched in which DB.

Thanks for the comment. We have included the following information on page 39:

The Literature Search

We conducted a Pubmed and Ovid Medline search of the literature from 2010 to 2022. The terms used covered a wide spectrum of terms associated with corneal imaging in infectious keratitis including: cornea, corneal ulcer, microbial keratitis, infectious keratitis, bacterial keratitis, fungal keratitis, HSV keratitis, viral keratitis, herpetic keratitis, optical coherence tomography, deep learning, convolutional neural networks, slit-lamp images, artificial intelligence, computer vision, machine learning; in vivo confocal microscopy. Reference lists from the recovered articles were also used to identify cases that may have not been included in our initial search.

  1. The development of AI related to the image of the external eye continues. A lot of literature is missing from the current review and needs to be added.

Thanks for the comment. However, this review focuses only on the corneal imaging for diagnosing infectious keratitis.

  1. It is also necessary to discuss the direction of AI development in the future. In particular, in the case of keratitis, AI should be produced using a small number of data. This approach has been introduced, for example, “Adopting low-shot deep learning for the detection of conjunctival melanoma using ocular surface images” and “Deep Sequential Feature Learning in Clinical Image Classification of Infectious Keratitis”.
  2. The authors should predict more about the future diagnosis of keratitis with the development of generative artificial intelligence and chatgpt. Please refer the following articles: “Artificial intelligence-based ChatGPT chatbot responses for patient and parent questions on vernal keratoconjunctivitis” and “New era after ChatGPT in ophthalmology: advances from data-based decision support to patient-centered generative artificial Intelligence”

Thanks for the above comments. We have added the section 5.3

5.3       Future perspectives.

Up to now, the majority of studies investigating the use of artificial intelligence in ophthalmology have focused on disease screening and diagnosis using existing clinical data and images based on machine learning and CNNs in conditions such as AMD, diabetic retinopathy, glaucoma and cataract (2). For infectious keratitis diagnosis, the generation of synthetic data using generative adversarial networks (GAN) may be a new method to train AI models without the need of thousands of images from real cases used in CNNs.  In case of less common conditions like fungal or acanthamoeba keratitis, a GAN could be utilised as a low-shot learning method via data augmentation meaning that conventional DL models could learn less common conditions using low number of images (3, 4). The low-shot learning technique had been used in detecting and classifying retinal diseases (5, 6) and in conjunctival melanoma (3).

Another AI technology that generates synthetic data is natural language processing (NLP) models such as ChatGPT developed by OpenAI (San Francisco, California) (2, 7). ChatGPT utilises DL methods to generate logical text based on the user’s ‘prompt’ in layman terms(8). ChatGPT was not conceived to specific tasks such as reading images or assessing medical notes, however, OpenAI had investigated the potential use of ChatGPT in healthcare and medical applications and research. Some applications include medical note taking and medical consultations. The medical knowledge embedded in ChatGPT may be utilised in tasks of medical consultation, diagnosis, and education with variable accuracy (9)For example, Delsoz et al. entered corneal medical cases (including infectious keratitis) on ChatGPT 4.0 and 3.5 to obtain a medical diagnosis which were compared with the results from three corneal specialists. The provisional diagnosis accuracy was 85% (17 of 20 cases) for ChatGPT-4.0 and 65% for ChatGPT-3.5 versus 100% (specialist 1), and 90% (specialist 2 and 3, each) (7). As a result, ChatGPT may be utilised to analyse clinical data along with DL models (CNNs or GAN) to diagnose and differentiate infectious keratitis.

  1. Early diagnosis and differential diagnosis of keratitis (bacterial, fungal, viral, or other inflammation) using AI are important problems. Emphasize this issue in introduction and conclusion.

Thanks for the comment.

Further information on early diagnosis and differential diagnosis of keratitis was added to the introduction and conclusion sections.

Reviewer 2 Report

The paper presents an important topic, and it is well-written.

Based on this impression I think the paper has the quality to be acceptable.

Minor changes will be welcome to further improve its presentations according to:

1) The style of the references doesn't match the journal template, please change as the example:

(1) would be [1]. Additionally, cross references will give a better way to check the references and Figures.

2) The reference should be presented before the point. Lines 40, 103. 

3) There are paragraphs that are too long, such as the second paragraph (Lines 41 to 61). 

Shorter paragraphs (from 5 to 10 lines) will make the paper easier to follow.

4) Some references don't match with the text. In line 125, the reference (7, 12, 20) should be only Abbouda et al. [20].

In line 134 the reference for the same author is not presented. Also, Konstantopoulos should be presented like this: Konstantopoulos et al. [12]

I think you could check all the references, and adjust it to have a better presentation.

5) I missed a section about the discussion. A table with the results will be more explainable than the text presented from lines 333 to 420.

6) A missed explanation about the architecture used in the convolutional neural network. The section about the method could be improved.

Follow a reference where you could find an example of the presentation that is missed: Video-Based Human Activity Recognition Using Deep Learning Approaches.

7) I think the conclusion should be divided, it is difficult to follow the ideas all grouped together.

It is acceptable as it is.

Author Response

Reviewer 2

1) The style of the references doesn't match the journal template, please change as the example: (1) would be [1]. Additionally, cross references will give a better way to check the references and Figures.

Thanks for the comment. Reference style was changed.

2) The reference should be presented before the point. Lines 40, 103. 

Thanks for the comment. This was amended.

3) There are paragraphs that are too long, such as the second paragraph (Lines 41 to 61). 

Shorter paragraphs (from 5 to 10 lines) will make the paper easier to follow.

Thanks for the comment. This was amended.

4) Some references don't match with the text. In line 125, the reference (7, 12, 20) should be only Abbouda et al. [20].

In line 134 the reference for the same author is not presented. Also, Konstantopoulos should be presented like this: Konstantopoulos et al. [12]

I think you could check all the references, and adjust it to have a better presentation.

Thanks for the comment. This section was removed

5) I missed a section about the discussion. A table with the results will be more explainable than the text presented from lines 333 to 420.

Thanks for the comment. Table 3 was added.

6) A missed explanation about the architecture used in the convolutional neural network. The section about the method could be improved.

Follow a reference where you could find an example of the presentation that is missed: Video-Based Human Activity Recognition Using Deep Learning Approaches.

Thanks for the comment.

Figure 8 was created based on a figure of a framework of several DL models to diagnose infectious keratitis in Kuo et al.

7) I think the conclusion should be divided, it is difficult to follow the ideas all grouped together.

Thanks for the comment. We have edited the conclusion section. See page 39.

  1. Conclusions

Infectious keratitis is among the top five leading causes of blindness overall worldwide. Early diagnosis of the causal organism is crucial to guide an adequate management to avoid severe complications such as vision impairment and blindness. Clinical examination under the slit-lamp is essential to diagnose the infection and the culture of corneal scrapes is still the gold standard in the identification and isolation of the causal organism.   Slit-lamp photography was revolutionised due to the invention of digital cameras and smartphones. Slit-lamp image quality essentially depends on the digital camera or smartphone camera sensor’s resolution, resolution of the slit-lamp and the focal length of the smartphone camera system. An image of 6 megapixels is sufficient in clinical photography. Alternative diagnostic tools using imaging devices such OCT and IVCM can be also used to track the progression of the infection and to determinate of the causal organism(s). The overall sensitivity and specificity of IVCM for fungal keratitis is 66-74% and 78-100%, and for Acanthamoeba keratitis is 80-100% and 84-100%.

 Due to the advancement of AI technology,  many study groups worldwide have evaluated the role of  DL-CNNs models to diagnose the different types of IK using either corneal photographs taken with smartphones or slit-lamp cameras or confocal images with variable promising results. Nevertheless, there are some challenges to overcome to use this technology in health settings including nonstandard image protocols, difficulty to find the common special features of the different types of IK, cost, accessibility, and cost-effectiveness. More recent DL models such as GAN may help in overcoming with these limitations generating synthetic data to train the models. Moreover, natural language processing models such as ChatGTP may analyse clinical data. These results along with GAN results may assist in diagnosing IK and differentiating the types of IK. Further work is needed to examine and validate the clinical performance of these CNNs models in the real-world healthcare settings with multiethnicity populations to increase the generalisability of the model. Finally, it would be also valuable to evaluate whether this technology assist in improving patient clinical outcomes via prospective clinical trials.

Reviewer 3 Report

The reviewed manuscript is a review of imaging methods for analysis of infectious keratitis. Various imaging techniques and the use of artificial intelligence methods for classification are discussed.

Comments:

1. General remark: The article is prepared quite haotically. The authors describe various solutions, but there are no critical comparisons, no indication of the advantages and disadvantages of individual solutions, and no discussion.

2. The title of the article is inadequate to its content. Two parts can be distinguished in the article: imaging techniques (OCT and IVCM) and the use of artificial intelligence. In my opinion, the title does not correspond to the content.

3. The Abstract should contain information about the content of the article. Most sentences from the Abstract should be moved to the Introduction.

4. Types of infectious keratitis should be clearly indicated in the Introduction.

5. Most of the solutions described in chapter 4 on AI are based on images from slit-lamps. It is difficult  to answer why only OCT and IVCM are described only, and the requirements for good imaging with slit-lamps are not described.

6. Chapter 2 contains the redundant subsection 2.2.1. TD OCT solutions are outdated and not currently used.

7. Figures 4 and 5 are of poor quality.

8. Figures 6 and 7 are disproportionately large (compared to Figures 4 and 5). Relevant areas of interest (ROI) should be indicated on these images.

9. Imaging methods should be summarized in a table, indicating the requirements, advantages, disadvantages, resolution of images and their dimensions, potential possibilities of detecting particular types of infectious keratitis.

10. Chapter 4 must be also supplemented by a comparative table (similar to that found in [41]). The table should include: author, year, dataset parameters, method, recognized classes, effectiveness (various metrics). On the basis of such a table, the advantages and disadvantages of individual solutions should be discussed.

11. Chapter 5 is based on [55] which is a major limitation.

12. A completed review paper should provide substantial new innovative ideas to the readers based on the comparison of published works. This has to be supplemented.

13. The Conclusions section should contain only relevant conclusions. The first three sentences should be deleted.

Minor notes:

14. Line 342: "camara"?

15. Line 426: "poor image quality" - too vague a term

16. Line 439: What does "global settings" mean?

In conclusion, the current version of the reviewed menuscript requires significant changes and additions beyond the "major revision".

Author Response

Reviewer 3

Comments:

  1. General remark: The article is prepared quite chaotically. The authors describe various solutions, but there are no critical comparisons, no indication of the advantages and disadvantages of individual solutions, and no discussion.

Thank you for the comment, the article has been further edited and a Table added to facilitate comparisons.

  1. The title of the article is inadequate to its content. Two parts can be distinguished in the article: imaging techniques (OCT and IVCM) and the use of artificial intelligence. In my opinion, the title does not correspond to the content.

The title has been changed to ‘Updates in diagnostic imaging for infectious keratitis” to reflect the use of AI as well as the latest imaging techniques.

  1. The Abstract should contain information about the content of the article. Most sentences from the Abstract should be moved to the Introduction.

Thanks for the comment.

We have edited the abstract:

Infectious keratitis (IK) is among the top 5 leading causes of blindness globally. Early diagnosis is key to guide an appropriate therapy to avoid complications such as vision impairment and blindness. Slit-lamp microscopy and culture of corneal scrapes are key   to diagnose IK. Slit-lamp photography was transformed when digital cameras and smartphones were invented. The digital camera or smartphone camera sensor’s resolution, resolution of the slit-lamp and the focal length of the smartphone camera system are key to a high-quality slit-lamp image. Alternative diagnostic tools include imaging such as with optical coherence tomography (OCT) and in vivo confocal microscopy (IVCM). OCT’s advantage is its ability to accurately determine the depth and extent of the corneal ulceration, infiltrates and haze; therefore, characterizing the severity and progression of the infection. However, it is not a preferred choice in the diagnostic tool package for infectious keratitis. IVCM is a great aid in the diagnosis of fungal and Acanthamoeba keratitis with overall sensitivities of 66-74% and 80-100%, and specificity of 78-100% and 84-100%, respectively. Recently, deep learning (DL) models has been shown to be promising aid in the diagnosis of IK via image recognition. Most of the studies that have developed DL models to diagnose the different types of IK have utilised slit-lamp photographs taken. Some studies have used extremely efficient single convolutional neural networks algorithms to train their models and others used ensemble approaches with variable results. This technology is likely to assist in the diagnosis of IK in some years; however, some limitations, including the need of large image datasets to train the models, the difficulty to find special features of the different types of IK, imbalance of training models, lack of image protocols and misclassification bias, need to be overcome to apply these models into real-world settings. Newer artificial intelligence technology that generates synthetic data such as generative adversarial networks may assist in overcoming some of these limitations of CNNs models.  

  1. Types of infectious keratitis should be clearly indicated in the Introduction.

Thanks for the comment. We have included two paragraphs with information about the types of infectious keratitis.

Infectious keratitis can be mainly caused by bacteria. Other important causal organisms include virus, fungi and parasites [4]. Bacterial keratitis is mostly caused by Staphylococci spp., Pseudomonas aeruginosa and Streptococcus pneumoniae. Patients generally manifest with a red eye, discharge, a corneal lesion, corneal infiltrates and sometimes hypopyon [9].  Viral keratitis is commonly caused by herpes simplex virus (HSV). The type of HSV keratitis is determined based on the clinical features observed on the slit-lamp examination. A dendritic or geographic ulcer is found in epithelial HSV keratitis. Stromal haze with or without ulcer, lipid keratopathy, stromal oedema, scarring, corneal thinning or vascularisation are found in stromal HSV keratitis. Stromal oedema and keratic precipitates are found in endothelial HSV keratitis. Stromal oedema, keratic precipitates and anterior chamber cells are found in HSV keratouveitis [10-12].

Fungal keratitis is caused by filamentous (Fusarium spp., Aspergillus spp.) or yeast (Candida spp.) fungi. Clinical findings include a corneal ulcer with gray or dirty-white surface with irregular feathery margins, elevated borders, or dry rough texture; satellite lesions, Descemet’s folds, hypopyon, ring infiltrate, endothelial plaque, anterior chamber cells and keratic precipitates [13-15]. Parasitic keratitis caused by Acanthamoeba spp. is an usual cause of infectious keratitis causing a chronic and progressing condition. A unilateral or paracentral corneal ulcer with a ring infiltrate is commonly seen in patients with this infection. At early presentation, patients may present with eyelid ptosis, conjunctival hyphemia, and pseudodendrites. At a later stage, deep stromal infiltrates, corneal perforation, satellite lesions, scleritis, and anterior uveitis with hypopyon may be found. Clinical symptoms include severe eye pain, decreased vision, foreign body sensation, photophobia, tearing and discharge [16-18]

  1. Most of the solutions described in chapter 4 on AI are based on images from slit-lamps. It is difficult  to answer why only OCT and IVCM are described only, and the requirements for good imaging with slit-lamps are not described.

Thank you for the comment a section on slit lamp imaging has been added.

  1. Slit-lamp biomicroscopy.

The slit-lamp is a stereoscopic biomicroscope which produces a focused beam of light with different height, width and angle to visualise and measure the anatomy of the adnexa and anterior segment of the eye [22]. This equipment is essential to examine and diagnose patients with infectious keratitis [4]. Attempts of slit-lamp photography started in late 1950’s; but the arrival of digital cameras in 2000’s substantially facilitated its use in ophthalmology [23]. There are two types of digital cameras: single lens reflex (SLR) or ‘point-and-shoot’. The choice of either for use in clinic will depend on the budget, easiness to use, photographic requirements and ability of the user. SLR are heavier, bulkier, and more costly than the ‘point-to-view cameras. A key feature to select a camera is the megapixel resolution. One megapixel is equivalent to one million pixels. A photograph taken at 6 megapixels can be printed up to 11 inches (28 centimetres) x 14 inches (35.5 centimetres) without ‘pixilation’ (visible pixels). Even a 3.2 megapixel camera suits the needs of clinical photography [24]. Other important features include macro mode for close-up photography of small objects; flash mode to light a dark scene or the object to be photographed,  additional flash fixtures to create diffuse illumination; ‘image stabilization’ or ‘vibration reduction’ technology to minimise camera shake to avoid blurred photographs [24].

An alternative to digital cameras is the smartphone which was released in the late 2000’s. A smartphone is a mobile phone with a cutting-edge technology to run many advanced applications in ophthalmology such as patient and physician education tools, testing tools, and photography [25]. Newer smartphones have rear camera resolution of up to 50 megapixels with image sensors, lens correction and optical plus electronic image stabilisation [26]. Due to the difficulty to hold the smartphone while operating the slit-lamp, adapters to mount smartphone have been developed [23]. However, there were some limitations to use the adapters including that they were specifically designed for certain smartphone or slit-lamp models, and when they were attached to the slit-lamp, binocular operation of the slit-lamp was not feasible.

To overcome these issues, Muth et al. evaluated a new adapter which can be mounted in any smartphone or slit-lamp and can be easily moved aside when binocular use is needed [23]. The images taken with the smartphone had an overall high quality and were as equally as good as the images taken with the slit-lamp camera [23]. Slit-lamp image quality depends on three factors: smartphone camera sensor’s resolution, resolution of the slit-lamp and the focal length of the smartphone camera system. The smartphone’s software setting including autofocus, shutter speed, and internal post-processing algorithms when using a compressed image format such as .jpg impact on the final image result. Newer smartphones that can take images in raw format will need more software-based post processing [23, 27].

  1. Chapter 2 contains the redundant subsection 2.2.1. TD OCT solutions are outdated and not currently used.

The section on TD OCT was removed.

  1. Figures 4 and 5 are of poor quality.

Thanks for the comment.

We are seeking the higher resolution versions of these photos.

  1. Figures 6 and 7 are disproportionately large (compared to Figures 4 and 5). Relevant areas of interest (ROI) should be indicated on these images.

Thanks for the comment.

We have added arrows to highlight the findings on the figures and a legend.

  1. Imaging methods should be summarized in a table, indicating the requirements, advantages, disadvantages, resolution of images and their dimensions, potential possibilities of detecting particular types of infectious keratitis.

Thanks for the comment. Table 1 was included.

  1. Chapter 4 must be also supplemented by a comparative table (similar to that found in [41]). The table should include: author, year, dataset parameters, method, recognized classes, effectiveness (various metrics). On the basis of such a table, the advantages and disadvantages of individual solutions should be discussed.

Thanks for the comment. Table 4 was added.

  1. Chapter 5 is based on [55] which is a major limitation.

Thanks for the comment. We have added some information and extra references. See page 38.

  1. A completed review paper should provide substantial new innovative ideas to the readers based on the comparison of published works. This has to be supplemented.

Thanks for the comments. We have added the section 5.3

5.3       Future perspectives.

Up to now, the majority of studies investigating the use of artificial intelligence in ophthalmology have focused on disease screening and diagnosis using existing clinical data and images based on machine learning and CNNs in conditions such as AMD, diabetic retinopathy, glaucoma and cataract [73]. For infectious keratitis diagnosis, the generation of synthetic data using generative adversarial networks (GAN) may be a new method to train AI models without the need of thousands of images from real cases used in CNNs.  In case of less common conditions like fungal or acanthamoeba keratitis, a GAN could be utilised as a low-shot learning method via data augmentation meaning that conventional DL models could learn less common conditions using low number of images [74, 75]. The low-shot learning technique had been used in detecting and classifying retinal diseases [76, 77] and in conjunctival melanoma [74].

Another AI technology that generates synthetic data is natural language processing (NLP) models such as ChatGPT developed by OpenAI (San Francisco, California) [73, 78]. ChatGPT utilises DL methods to generate logical text based on the user’s ‘prompt’ in layman terms[79]. ChatGPT was not conceived to specific tasks such as reading images or assessing medical notes, however, OpenAI had investigated the potential use of ChatGPT in healthcare and medical applications and research. Some applications include medical note taking and medical consultations. The medical knowledge embedded in ChatGPT may be utilised in tasks of medical consultation, diagnosis, and education with variable accuracy [80]. For example, Delsoz et al. entered corneal medical cases (including infectious keratitis) on ChatGPT 4.0 and 3.5 to obtain a medical diagnosis which were compared with the results from three corneal specialists. The provisional diagnosis accuracy was 85% (17 of 20 cases) for ChatGPT-4.0 and 65% for ChatGPT-3.5 versus 100% (specialist 1), and 90% (specialist 2 and 3, each) [78]. As a result, ChatGPT may be utilised to analyse clinical data along with DL models (CNNs or GAN) to diagnose and differentiate infectious keratitis.

  1. The Conclusions section should contain only relevant conclusions. The first three sentences should be deleted.

Thanks for the comment. We have edited the conclusion section.

Conclusions

Infectious keratitis is among the top five leading causes of blindness overall worldwide. Early diagnosis of the causal organism is crucial to guide an adequate management to avoid severe complications such as vision impairment and blindness. Clinical examination under the slit-lamp is essential to diagnose the infection and the culture of corneal scrapes is still the gold standard in the identification and isolation of the causal organism.   Slit-lamp photography was revolutionised due to the invention of digital cameras and smartphones. Slit-lamp image quality essentially depends on the digital camera or smartphone camera sensor’s resolution, resolution of the slit-lamp and the focal length of the smartphone camera system. An image of 6 megapixels is sufficient in clinical photography. Alternative diagnostic tools using imaging devices such OCT and IVCM can be also used to track the progression of the infection and to determinate of the causal organism(s). The overall sensitivity and specificity of IVCM for fungal keratitis is 66-74% and 78-100%, and for Acanthamoeba keratitis is 80-100% and 84-100%.

 Due to the advancement of AI technology, many study groups worldwide have evaluated the role of DL-CNNs models to diagnose the different types of IK using either corneal photographs taken with smartphones or slit-lamp cameras or confocal images with variable promising results. Nevertheless, there are some challenges to overcome to use this technology in health settings including nonstandard image protocols, difficulty to find the common special features of the different types of IK, cost, accessibility, and cost-effectiveness. More recent DL models such as GAN may help in overcoming with these limitations generating synthetic data to train the models. Moreover, natural language processing models such as ChatGTP may analyse clinical data. These results along with GAN results may assist in diagnosing IK and differentiating the types of IK. Further work is needed to examine and validate the clinical performance of these CNNs models in the real-world healthcare settings with multiethnicity populations to increase the generalisability of the model. Finally, it would be also valuable to evaluate whether this technology assist in improving patient clinical outcomes via prospective clinical trials.

Minor notes:

  1. Line 342: "camara"? Amended.
  2. Line 426: "poor image quality" - too vague a term

This sentence was amended.

  1. Line 439: What does "global settings" mean?

This sentence was amended.

Round 2

Reviewer 1 Report

The manuscript has been improved to describe the literature, but should further be edited to improve the readability.

Author Response

Thanks for the comment.

We have improved the readability of the text.

Reviewer 3 Report

The article has been corrected and new tables have appeared. However, the article requires corrections and additions.

Remarks:

1. The authors supplemented the article with types of keratitis. However, a diagram with division would be useful.

2. Figures 1, 2,3: There is no information about where the images come from.

3. Figures 4 and 5 are still of poor quality.

4. Tables 1 and 4 are difficult to read. Their arrangement should be changed and the solutions with the best effectiveness should be marked.

5. Some sentences are taken out of context. For example, line 256 "Twelve patients required tarsorrhaphy or corneal gluing; six, deep anterior lamellar keratoplasty (DALK), and one, vitrectomy. The average score of the surgical patients was 19." This paragraph requires a meaningful introduction.
